# The effect of training doctors in communication skills on women's satisfaction with doctor–woman relationship during labour and delivery: a stepped wedge cluster randomised trial in Damascus

Hyam N Bashour,[1] Mona Kanaan,[2] Mayada H Kharouf,[1] Asmaa A Abdulsalam,[3] Mohammed A Tabbaa,[3] Salah A Cheikha[3]

[1]Department of Family and Community Medicine, Damascus University, Damascus, Syria
[2]Department of Health Sciences, University of York, York, UK
[3]Departmentof Obstetrics and Gynaecology, Faculty of Medicine, Damascus University, Damascus, Syria

Correspondence to
Dr Hyam N Bashour;
hbashour@scs-net.org

## ABSTRACT

**Objectives:** To determine the effect of training residents in interpersonal and communication skills on women's satisfaction with doctor–woman relationship in labour and delivery rooms.

**Design:** A stepped wedge cluster randomised trial.

**Setting:** 4 tertiary care teaching maternity hospitals in Damascus, Syria.

**Participants:** 2000 women who gave birth to a living baby in the four study hospitals and consented to participate in the intervention took part in the study. Women with difficult labour and high-risk pregnancies were excluded. All were interviewed at home after discharge.

**Interventions:** A specially designed training package in communication skills was delivered to all resident doctors at the four hospitals.

**Primary outcome measures:** The main outcome measure was women's satisfaction with interpersonal relationships in labour and delivery rooms measured via a series of questions on a Likert scale modified from the Medical Interview Satisfaction Scale.

**Results:** At the individual level, the mean for the average satisfaction score was 3.23 (SD 0.72) of a possible score of 5 in the control group and 3.42 (SD 0.73) in the intervention group. Using generalised linear mixed models, we were unable to detect a difference between the mean for the average satisfaction score of women in the intervention arm and that of women in the control arm; the 95% CI associated with the effect of the intervention ranged from −0.08 to 0.15.

**Conclusions:** Despite slight changes in the observed residents' communication skills, the training package in communication skills does not seem to be associated with higher satisfaction scores of women. This raises the question of whether training individuals without further structural changes in the delivery of care and without further reinforcement of the training can have an impact on improving the quality of doctor–patient communication.

**Trial Registration Number:** ISRCTN80243969

## ARTICLE SUMMARY

### Article focus
- The study aims to determine the effect of training residents in interpersonal and communication skills on women's satisfaction with the doctor–woman relationship in labour and delivery rooms.
- The context of the trial focuses on teaching maternity hospitals with high patient volume.
- There have been few randomised controlled trials testing interventions to improve maternal healthcare in the Arab region and in Syria in particular.

### Key messages
- In the context of a highly crowded and stressful environment where middle-class and low-class Syrian women give birth, a specially designed training package in interpersonal and communication skills for residents did not achieve an overall improvement in women's satisfaction with the doctor–woman relationship in labour and delivery.
- This study raises questions as to whether training individuals without more structural changes in the delivery of care can have an impact on improving the quality of doctor–patient communication. These are important questions to address and call for further research.
- Despite the lack of evidence from this study, the need to improve the interpersonal skills of medical doctors and obstetricians specifically should be reinforced, as good communication is central to quality healthcare.

## BACKGROUND

There is a growing body of literature linking health providers' communication skills to a host of valued patient outcomes, including satisfaction, adherence and positive health

## ARTICLE SUMMARY

### Strengths and limitations of this study
- The key strength of the study was in its stepped wedge cluster randomised design, which is known to be ethical and practical. The primary limitation of the study was its inability to link women's satisfaction, or dissatisfaction, with the behaviour of a specific doctor or specific doctors; for ethical and practical reasons, we observed actions rather than the behaviours of individual residents. This was unavoidable due to the design's key considerations.

indicators.[1] As a result, communication in healthcare, particularly between healthcare professionals and patients, has attracted an increasing amount of attention at official and professional levels in recent years. Many academic and statutory bodies have been involved in initiatives to promote good communication including that for maternity care.[2 3] These initiatives are, in part, a response to increasing evidence from researchers that the quality of the interaction between patients and their care providers may have a significant effect on a variety of aspects of patient's wellbeing. These include satisfaction, knowledge and understanding, adherence with advice or treatment, quality of life, and psychological and other health outcomes.[4 5]

Maternity care is an area of healthcare in which importance of good communication has received particular attention.[3] Studies of women's views of maternity care suggest that good communication is central in determining whether women are satisfied with the care that they receive or not; communication gives them the opportunity to be better informed and ask questions, and it signals understanding and respect.[6 7] Observational studies of communication between midwives and women during labour have highlighted the misunderstandings that can occur when communication is poor and have identified areas where communication could be improved.[8]

Despite the acknowledged importance of communication in maternity care and the official recognition that communication is not always as good as it should be, there have been few evaluations of strategies to improve communication between women and their care providers. Previous reviews of doctor–patient communication have not included any studies carried out in maternity care.[4 5] The review carried out by Rowe et al[9] aimed to fill in that gap and thus identified and reviewed trials of the effectiveness of interventions aimed at improving communication between health professionals and women in maternity care. The review identified trials largely in the area of antenatal care but also identified a major gap in knowledge relating to communication in several key areas of labour, delivery and postnatal care. They recommended that trials of interventions to improve communication between carers and women in labour and in the postnatal period would be particularly useful. All the studies included in that review were carried out in developed countries, mainly in the UK.

Although good progress in reducing maternal mortality has been achieved in Syria, maternity care is still characterised by fragmentation of care, lack of protocols including those related to pain relief in labour, overmedicalisation of care, strong and even informal role of private sector, complex relationships between skilled attendants and variation of quality of care received.[10]

In Syria, previous work at both public hospitals and community level showed that continuous support of women during labour and delivery is virtually nonexisting. Companionship is not allowed at hospitals, however, in a population-based study women reported the existence of a companion at delivery in 37% of the cases.[11] Syrian women expressed their dissatisfaction at being left alone and attended by care providers who lack good communication skills. It was evident that Syrian women taking part in the study were left without any social and emotional support during a critical period in their lives. In most occasions, they were not allowed to be accompanied by their relatives when in labour and delivery. There were many instances in which doctors and midwives themselves were the source of dissatisfaction due to lack of respect for patients and discrimination which is mainly related to the patient's socioeconomic status.[12]

Furthermore, labour and delivery care is largely provided in overcrowded hospitals mainly by medical graduates with little or no training in communication and interpersonal skills and where midwives and nurses' roles are typically marginal. These poor conditions might well impact women's satisfaction in the delivery process. This is added to the fact that Syrian public hospitals do not have a policy to administer pain relief; the low societal status of women especially those from lower socioeconomic backgrounds and the restrictive relations between women and men outside the family context. For example, previous research findings showed that in the delivery suites, eye-to-eye contact was not acceptable if the gender of the service provider was male.[12]

Therefore, the main objective of this trial was to evaluate the effect of training resident doctors, as the main providers of care in teaching public maternity hospitals, in interpersonal and communication skills on women's satisfaction with doctor–woman relationship in labour and delivery rooms using a modified version of the Medical Interview Satisfaction Scale (MISS-21).[13] The secondary objective of this study was to measure the change in communicative behaviour of residents at labour and delivery rooms level using items of the Al-Galaa observational checklist.[14]

## METHODS
### Study design
A stepped wedge cluster randomised design was used in this trial.[15 16] This particular design as reviewed by Brown and Lilford[16] involves sequential roll-out of the intervention to individuals or clusters over a number of time periods but the order in which participants receive the intervention is determined at random. In this trial,

the roll-out of the training package was carried out at four time points separated by 2 months each in addition to a baseline time point (see supporting file one for a diagram of the trial design).

At baseline, each hospital contributed a cluster to the control arm whereas at the last time period each hospital contributed a cluster to the intervention arm. The timing of the implementation of the training package in each hospital determined when clusters switched from the control arm to the intervention arm. A coded list of the hospitals was produced and a statistician blinded to the coding allocated randomly the timing of the introduction of the intervention to each hospital. By the end of the study, all eligible resident doctors in the study hospitals received the intervention.

## Participants

Participants in this study included care providers and women delivering at four public maternity hospitals in Damascus and its surroundings, Syria. The study hospitals included three hospitals affiliated to the Ministry of Health (MOH) and one university teaching hospital. The hospitals were the Maternity Teaching hospital at Damascus University (15 000 deliveries each year), Al-Zahrawi hospital, which is the largest MOH hospital in Damascus (13 000 deliveries each year), Douma hospital, which is an MOH district hospital in rural Damascus (7000 deliveries/year) and Harasta hospital, an MOH district hospital in rural Damascus (5000 deliveries/year).[17]

Care providers to undertake the training intervention were all residents registered for the year of field work (2008–2009) in the four hospitals. Midwives were excluded from this study because they had no role in the delivery process, except for assisting doctors, according to policies at those hospitals.

Women were included if they had normal vaginal birth or by caesarean section and gave birth to a living baby at the four hospitals during the study period. Women using the study public hospitals come largely from middle and low socioeconomic backgrounds.

Informed consents were obtained from residents and women participating in the study. The study protocol was approved by the Institutional Review Board at Damascus University. All study hospitals agreed to take part in this research.

## Intervention

The intervention consisted of exposing all residents in the four study hospitals to a training package in interpersonal and communication skills, using a specially designed training package that has been developed by the study team in cooperation with an international expert in the field. The objectives of the training package were as follows: to recognise the impact of effective communication on women and child health during labour and delivery; to identify characteristics and principles of effective communication; to recognise and be able to overcome barriers to effective

communication; to enhance and reinforce the interpersonal communication skills of health providers and to improve their interactions with patients in general and women in labour in particular.

The content of the package included an evidence from a labour room. Issues regarding attitudes of health providers, overview of doctor–woman communication, non-verbal communication, building rapport, listening skills, effective interviewing, counselling and persuasion and the ideal maternity ward were all explored. The methodology of the workshop was based on sharing concepts and ideas; self-assessment by means of tests and checklists; brainstorming, group work and plenary discussion; learning by experience; role-play; games and ice breakers (the training package is available for readers on request).

All residents at the four hospitals received the intervention; their total number was 137. Training was carried out by a national trainer with experience in communication skills together with members of the research team who observed and facilitated the training. In total, nine training workshops were conducted at the Faculty of Medicine, Damascus University. The duration of each workshop was 20 h in total, delivered over 3 days. The rate of daily attendance ranged from 78% to 100%. A formal evaluation of the training workshops was carried out. The feedback from the evaluation was very positive with 97% of respondents saying that they would recommend the workshop. However, 82% envisaged barriers to implementation such as time pressure, overloaded work and hospital's routine.

## Data collection

Data on women's satisfaction were collected prior to the intervention and at each instance of the four randomisation points from all hospitals. Data collection took place between April 2008 and January 2009. Participating women were interviewed at their homes within 2 weeks after delivery. Data were collected on demographic and socioeconomic variables (age, education, work status, husband's age, husband's education, husband's occupation and home ownership), and about the pregnancy and delivery (whether the mother is multiparous/ nulliparous, gender of the newborn, type of delivery), also women were asked to complete a modified version of MISS-21.[13]

Furthermore, at each randomisation point data were collected to describe the communicative behaviour of care providers. Trained observers were asked to fill in a checklist recording observations of collective communicative behaviour of residents at the labour and delivery rooms rather than the individual residents throughout the different hospital shifts using the Al-Galaa checklist.[14] Three shifts were observed per day which amounted to a total of 24 h observation of each delivery room at each point of time. We have chosen this approach as we were more interested in the change of behaviour at the service level rather than the actual

change in behaviour of an individual resident. Furthermore, this approach was expected to reduce problems that could arise if residents were observed individually. Thus, scores from observation were collected at the level of the shift's round rather than scores for each resident. Pretraining and post-training measurements were carried out in all four hospitals at each point of time. Thus, the observation was carried out on an average of 2–3 weeks after the implementation of the training package in each hospital, with some variance between large and small hospitals. The training took longer time at large hospitals because of the nature of the workshop and the necessity to implement it two to three times to cover all residents. Observers and field workers were blind to the specific objectives of the study.

Qualitative data collection using six focus group discussions also took place prior to the field work to serve the design phase of this project. The qualitative methods aimed to understand the work environment of the residents and get their views concerning the importance of designing a training package in communication skills.

### Outcomes

The primary outcome of this study was women's satisfaction with interpersonal and communication skills of doctors working in labour and delivery rooms, which could be seen as a patient outcome in the Ong et al's[4] model. The theoretical model for doctor–patient/woman communication as suggested by Ong et al (1995) is a useful model to conceptualise an interventional research that aims to improve communication. It consists of background variables, actual content of communication as process indicators as well as patient outcomes including satisfaction in the short time. Women's satisfaction was measured using a Likert scale questionnaire investigating the communication skills of residents attending her. The questionnaire was based on the patient satisfaction questionnaire MISS-21;[13] [18] the adapted questionnaire is referred to as the Modified Medical Interview Satisfaction Scale (MMISS). The validation exercise of our measurement tool which was implemented in Arabic language is to be reported elsewhere. The questionnaire had a total of 21 questions focusing on the communication skills of the doctors in our setup. Women were asked to indicate their levels of satisfaction with communicative practices of the attending resident on a 5-point Likert scale with 1 indicating strongly disagree to 5 indicating strongly agree. There were eight negatively worded questions; the codes of these questions were reversed for analyses purposes. A score was then calculated by summing up the women's answers to the MMISS questions except for the first one, which was a general satisfaction question. Therefore, the highest possible score was 100 and the lowest was 20; the higher values indicated that women were satisfied with the services provided to them. An average score was then calculated by dividing the score by the number of questions.

Two scenarios were investigated. In the first scenario, the average score was computed for participants who provided information for the 20 items on the MMISS questionnaire and is referred to as the average score for complete MMISS. In the second scenario, the average score was computed for items with a response and is referred to as the average score only.

The secondary outcome measured the communicative behaviour of residents serving in labour and delivery rooms using sections related to communication of the observational checklist developed by Al-Galaa study in Egypt by Sholkamy et al[14]. The Al-Galaa study checklist was developed to document normal labour and delivery practices in an Egyptian hospital and covered areas such as management of labour, communication, postpartum and neonatal care; the instrument was tested and validated.[14] The checklist could be considered as a process evaluation in this work. It constituted 31 questions and included items such as whether the attending doctor greeted the woman, identified himself/herself, explained the procedures of the medical examination he/she was about to undertake and communicated the findings, responded to the woman's questions, gave clear instructions about the different stages of labour/delivery and what was expected from the woman and whether the doctor gave any guidance with regard to post-delivery.

### Sample size

Power calculations were carried out as in Hussey and Hughes[15]. A pilot study of 10 women resulted in a mean score of 3.165 (63.3 points on the satisfaction scale) and a SD of 0.71. Based on a cluster size of a 100 at each step and hospital, a mean of 3.15 (63 points) and an SD of 0.75 points, a difference of 0.2 (4 points) could be detected with 90% power given a significance level of 0.05. Therefore, the total sample size needed for the study was 2000 women (100 women×4 hospitals × 5 time points) not accounting for non-response rate.

The total number of all residents in the four hospitals was 137. They were distributed as follows: 85 in the Maternity Teaching Hospital; 31 in Al-Zahrawi Hospital; 7 in Douma Hospital and 14 in Harasta Hospital. They all received the intervention when it was offered at their respective hospitals.

### Statistical methods

Baseline characteristics of the women were summarised using counts (percentages) for categorical variables and means (SD) for continuous variables.

Statistical analysis was based on the principle of intention to treat. Comparison of the main and secondary outcomes between the control and intervention arms of the study was carried out at the individual level. A Generalised Linear Mixed Model (GLMM) was used to determine the size and direction of the difference between the control and intervention arms for the main and secondary outcomes. Estimates of the difference

and 95% CIs were calculated. All analyses were adjusted for clustering and time of the intervention. Additional analyses of the main outcome were conducted controlling for demographic variables. Furthermore, we used multiple imputations to address missing data.

Statistical analyses were carried out using STATA software (College Station, Texas, USA) and the R language (http://www.r-project.org).

## RESULTS

A total of 2000 women delivering in the four hospitals were assessed for their satisfaction with doctor–woman relationship. Figure 1 describes the flow of participants using a CONSORT diagram.[19] All women approached agreed to participate in the trial.

Table 1 shows the characteristics of the women by treatment arm. The mean age of the women was 25 years (SD: 6 years). Approximately 95% were housewives. Three quarters had primary education only, 13% were illiterate whereas 12% had high school or further education qualifications. However, there was a differential in the distribution between hospitals with the per cent illiterate ranging from 3.6% to 23%. This reflected the fact that two of the hospitals served an urban population, whereas the other two served a rural population.

The husband's mean age was 31 years (SD: 6.76 years). The vast majority of husbands were self-employed (41%), followed by being a manual worker (35%) and clerk (18%). The distribution of the husband's education was similar to that of the woman's education. Almost half owned their house, whereas the other half lived in shared accommodation. A quarter of the women were nulliparous and only two women had a C-section.

The percentage of women agreed or strongly agreed with the statement of overall satisfaction with doctors' communication skills during labour/delivery ranged from 51% to 83% between hospitals in the control arm and 58% to 85% between hospitals in the intervention arm. These percentages were higher in the intervention arm for all hospitals except for Hospital 2 which decreased from 83% to 67% in the intervention arm. For presentation purposes, the hospitals were coded such that Hospital 1 represented the hospital that had the intervention delivered first and so forth.

Table 2 details the views of women on specific questions with regard to doctors' communication skills. The majority of women pointed-out that doctors did not identify themselves prior to the medical examination; the percentages were similar between the control and intervention arm. One-third of the women in the intervention arm agreed/strongly agreed that the doctors'

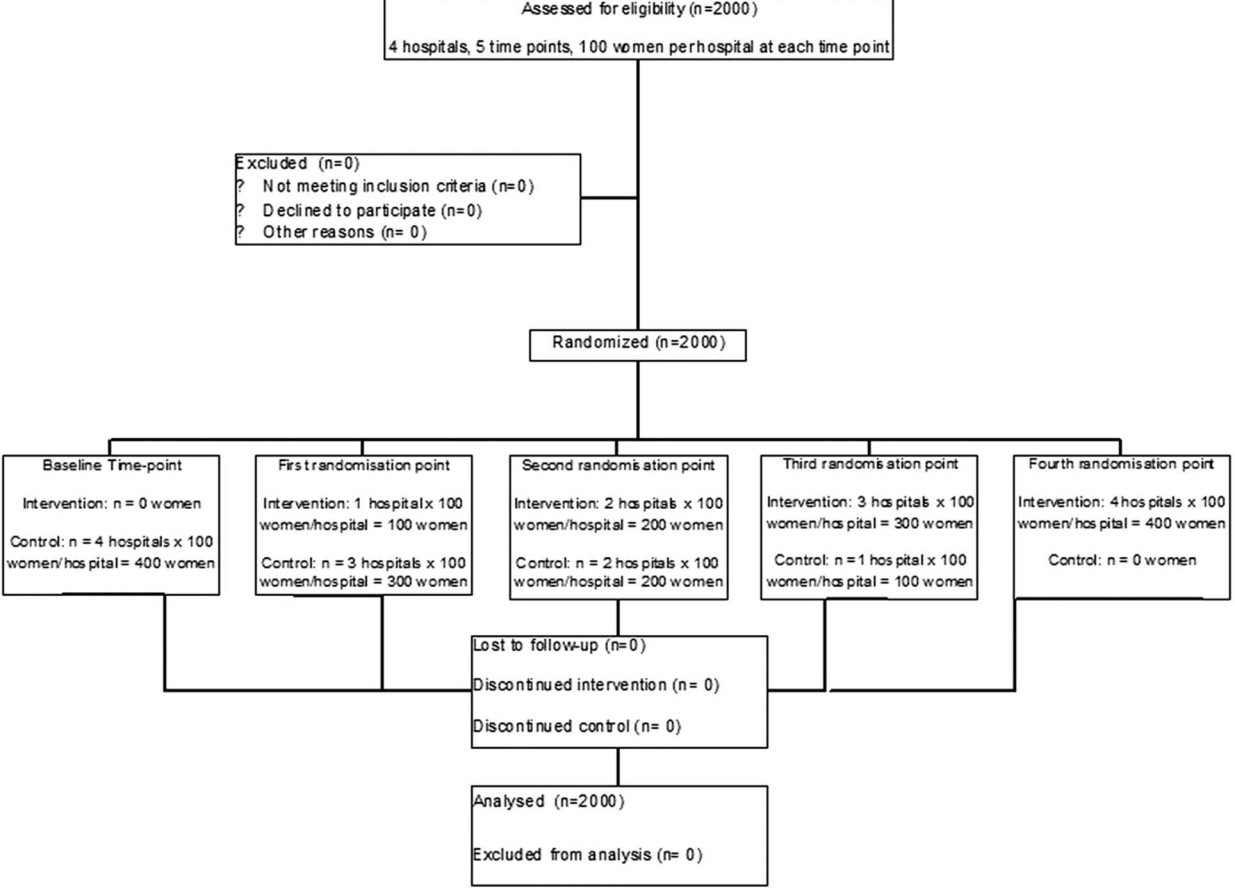

**Figure 1** Flow of participants.

**Table 1** Demographic characteristics of the women by intervention arm

| | Control N=1000 | Intervention N=1000 | | Control N=1000 | Intervention N=1000 |
|---|---|---|---|---|---|
| Mean age (SD) | 25.02 (6.11) | 25.10 (5.89) | Woman's education | | |
| Occupation: housewife | 949 (95) | 953 (95) | Illiterate/read and write | 140 (14) | 102 (10) |
| Home ownership: share | 488 (49) | 470 (48) | Primary | 745 (74) | 761 (77) |
| Parity: nulliparous | 266 (27) | 263 (27) | High school + | 110 (11) | 128 (13) |
| New born gender: male | 507 (52) | 480 (49) | Husband's education | | |
| Mean husband's age (SD) | 30.90 (6.91) | 30.85 (6.62) | Illiterate/read and write | 113 (11) | 101 (10) |
| Husband's occupation | | | Primary | 781 (79) | 743 (76) |
| Labourer (manual worker) | 359 (36) | 337 (34) | High school + | 100 (10) | 135 (14) |
| Clerk | 160 (16) | 198 (20) | | | |
| Self-employed | 391 (40) | 432 (44) | | | |
| Other | 80 (8) | 21 (2) | | | |

Numbers represent counts (percentage) unless otherwise stated.

greeted them at the onset of the consultation compared with a fifth in the control arm. Around 40% of the women agreed/strongly agreed that the doctor looked at them when he/she talked to the woman in the control arm compared with 60% in intervention arm. Similar percentages were observed for whether the

**Table 2** Numbers (percentages) of women who agreed/strongly agreed with various questions concerning their labour/delivery experience

| Item | Control period N=1000 | Intervention period N=1000 | Item | Control period N=1000 | Intervention period N=1000 |
|---|---|---|---|---|---|
| Overall satisfaction with doctors' communication skills during labour/delivery | 619 (62) | 713 (72) | Did the doctor listen to you with concern and without interrupting you? | 413 (46) | 551 (58) |
| Did the doctors identify themselves? | 132 (13) | 118 (12) | Was the doctor engaged in other issues, so you felt being unattended and annoyed? | 165 (17) | 67 (7) |
| Did the doctor greet you? | 208 (21) | 323 (32) | Did you feel let down by the way he/she was dealing with you? | 66 (7) | 31 (3) |
| Did the doctor look at you when talking to you? | 434 (43) | 599 (60) | Did the doctor explain all the steps before doing the clinical exam? | 249 (25) | 278 (29) |
| Did the doctor show interest in you as a person? | 393 (39) | 329 (33) | Did the doctor explain the findings from the clinical exam? | 326 (34) | 399 (41) |
| Did the doctor insult you? | 55 (6) | 29 (3) | Were you annoyed by the doctor explaining all the findings? | 40 (5) | 73 (10) |
| Did the doctor use humour to comfort you? | 78 (8) | 118 (12) | Did the doctor explain all the alternative choices to you? | 176 (23) | 145 (17) |
| Did the doctor use any terms to calm you down? | 363 (36) | 419 (42) | Did the doctor help you in making decisions? | 175 (29) | 158 (23) |
| Did the doctor use his/her hands to assist you or to reassure you? | 314 (32) | 409 (42) | Was the doctor annoyed from your questions and avoided answering them? | 122 (14) | 54 (6) |
| Did the doctor shout or scream at you? | 125 (13) | 71 (7) | Did the doctor use medical terms in explanation, thus they were not understood? | 77 (8) | 48 (5) |

The numbers compare all the intervention clusters to the control clusters, due to missing information some of the percentages do not tie up with the total number given.

doctor listened to the woman with concern and without interruption. Almost 40% of the women agreed/strongly agreed that the doctor showed an interest in them in the control arm versus a third in the intervention arm. Forty-two per cent of the women in the intervention arm agreed/strongly agreed that the doctor used some terms to calm them down and used his/her hands to assist/reassure them compared with 36% and 32% in the control arm, respectively. A small percentage reported that the doctor insulted them (6% control arm vs 3% intervention arm) or shouted/screamed at them (13% control arm vs 7% intervention arm). Only a quarter of the women agreed/strongly agreed that the doctor explained the examination steps before the clinical examinations in the control arm compared with 29% in the intervention arm. A third agreed/strongly agreed that the doctor explained the results from the clinical examination in the control arm compared with a 41% in the intervention arm, however, Hospitals 1 and 2 had a decrease in the percentages who agreed/strongly agreed, whereas Hospitals 3 and 4 had an increase. Only a quarter of those who responded to the questions whether the doctors explained alternative choices or helped the woman in making decisions agreed/strongly agreed to these statements.

Table 3 provides summary statistics for the intervention and control arm for the two outcome scores: average score for MMISS and average score for complete MMISS. The median and IQR (Q1, Q3) score for the average score was 3.3 (2.8, 3.7) in the control arm and 3.3 (3.0, 4.1) in the intervention arm; see online supplementary appendix for a graphical representation of the average score. Furthermore, the median and IQR score for the average score of the complete MMISS was 3.4 (3.0, 3.9) in the control arm and 3.3 (3.1, 4.1) in the intervention arm (see additional file 2 for a graphical representation of the average score for complete MMISS; figure 2).

Regression estimates for the average score on intervention and time using GLMM are given in table 3. The effect of the intervention was an average increase of 0.03 points; however, there was no evidence for statistical significance (95% CI: −0.08 to 0.15). The estimates using complete MMISS implied an average decrease of 0.13 in the intervention arm compared with the control arm (equivalent to 2.6 points on the original scale). Nevertheless, the effect was not statistically significant with a 95% CI (−0.29 to 0.04). We also carried additional analyses adjusting for the woman's demographic characteristics. The results of the intervention effect were similar to the main analysis and none of the demographic variables was a significant predictor. We also used multiple imputations to account for missing data. The estimates of this analysis (data not shown) were similar to those reported here.

Table 4 reports on the secondary outcome, the observational checklist and provides the numbers and percentages of when an item on the observational checklist, concerning communication between doctor and woman during labour/delivery, was not observed. Based on

**Table 3** Summary statistics for the outcome of women satisfaction computed as the average score and average score for complete MMISS

| | Average score | Average score complete MMISS |
|---|---|---|
| Control: median (Q1, Q3) | 3.3 (2.8, 3.7) | 3.4 (3.0, 3.9) |
| Intervention: median (Q1, Q3) | 3.3 (3.0, 4.1) | 3.3 (3.1, 4.1) |
| GLMM estimates *Fixed-effects* | $_{Lower}Est_{Upper}$ | $_{Lower}Est_{Upper}$ |
| Intervention | $_{-0.08}0.03_{0.15}$ | $_{-0.29}\text{-}0.13_{0.04}$ |
| Time: 2 | $_{-0.08}0.02_{0.12}$ | $_{-0.10}0.05_{0.19}$ |
| Time: 3 | $_{-0.14}\text{-}0.03_{0.08}$ | $_{-0.02}0.12_{0.27}$ |
| Time: 4 | $_{-0.08}0.05_{0.18}$ | $_{0.12}0.30_{0.49}$ |
| Time: 5 | $_{-0.15}0.01_{0.16}$ | $_{-0.03}0.17_{0.37}$ |
| Constant | $_{3.16}3.32_{3.48}$ | $_{3.23}3.44_{3.64}$ |
| *Random-effects* | Variance (SE) | Variance (SE) |
| *Hospital* | 0. 02 (0.016) | 0.03 (0.026) |
| Residual | 0.51 (0.016) | 0.45 (0.02) |

Regression analysis estimates for the average score and average score for complete MMISS on intervention and time controlling for clustering by hospital using GLMM. Estimates are presented together with 95% CI limits as subscripts ($_{Lower}Est_{Upper}$). GLMM, Generalised Linear Mixed Model; MMISS, Modified Medical Interview Satisfaction Scale.

table 4, the majority of doctors did not identify themselves, did not explain the steps of labour and delivery, did not explain to the woman the stages of labour and her role in it, and did not give any instructions about the steps that follow delivery or how to care for her new born. In 9 of the 31 items, the proportion of negative responses was consistently lower in the intervention arm compared with the control arm across the hospitals; data shown in supplementary file. In the intervention arm the proportion of doctors who called the woman by her name or title increased from 75% to 85%, those who asked for permission to examine the woman from 39% to 51%, and before the vaginal examination from 38% to 58% compared with the control arm. The practice of covering the woman during delivery increased in the intervention arm to 81% from 41% though there were variations across hospitals; asking the woman to bend her knees increased to 85% from 61%; relaying information to other doctors in the team increased to 83% from 54% and congratulated the woman on her delivery increased to 60% from 47% in the control arm.

Using GLMM regression, scores of the observational checklist at the shift level were compared pre- and post-intervention adjusting for clustering at the hospital level. This showed no evidence of statistical significance with an estimate of −0.01 (95% CI −0.03 to 0.02) for the intervention parameter.

## DISCUSSION

Maternity care is an area of healthcare in which the importance of good communication has received

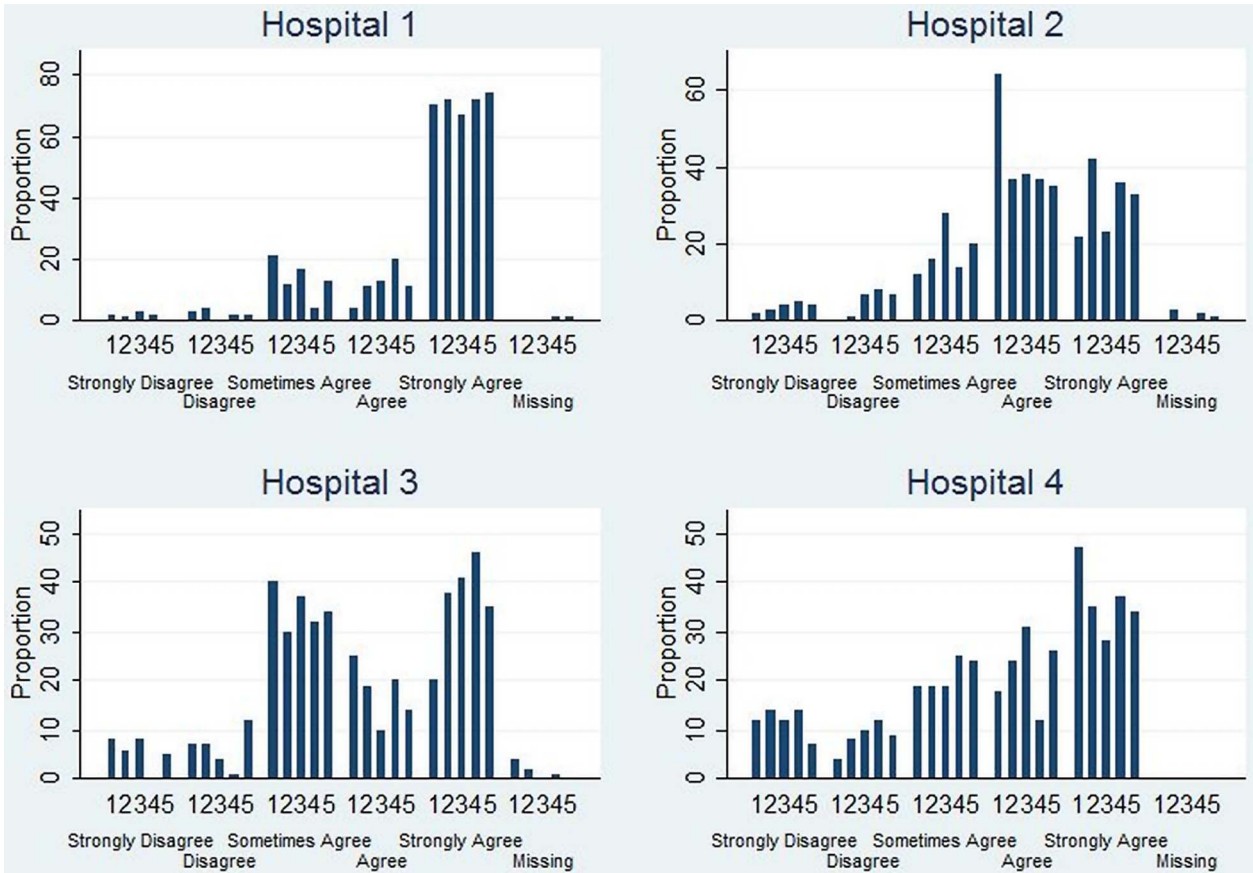

**Figure 2** A graphical representation of the overall satisfaction question of the women stratified by hospital. Data presented represent proportion in each category of overall satisfaction at the different time periods.

particular attention, as good communication skills are essential for high quality, effective and safe medical practice.[1–3] This trial was designed to determine the effectiveness of training residents at maternity wards in interpersonal and communication skills on women's satisfaction with doctor–woman relationship in labour and delivery rooms in four teaching hospitals in Damascus. Syrian women are largely delivered by medical graduates with little or no training in communication and interpersonal skills. Results of this trial did not provide evidence that the intervention improved women's satisfaction as measured by a modified version of the MISS-21. Communicative behaviour of the residents measured by an observational checklist was poor before the intervention and showed a slight improvement although not statistically significant after training on some selected items such as asking the woman's permission for the examination or asking her to bend her knees. Violent behaviours were also noted and reported such as screaming and shouting.

The reasons for those negative findings of our trial could be multifaceted. First, we have no reason to think that the negative findings are due to measurement errors having validated the tool and trained the field workers. However, the large numbers of care providers at maternity wards including midwives, nurses and junior doctors and even cleaners in the wards could have played a role on the overall satisfaction of women, given that midwives and nurses were not part of our target group for training as their role is mainly to assist doctors rather than taking a full charge of delivery. Furthermore, the policy implemented at Syrian public hospitals of not administering pain relief might have also confound the impact of the intervention on satisfaction as women would definitely prefer it.[11] The training package was tailored to our setting and was informed by previous research findings; it allowed for Ong's useful model to conceptualise an interventional research that aimed to improve communication and it ensured that the key tasks in communication are covered using effective teaching methods.[4 20] However, lack of opportunity to reinforce the training could have played a negative role. Brown et al[21] suggested that communication skill programmes may need to be longer, more intensive, teach a broader range of skills, provide ongoing performance feedback, utilise patient feedback and use a variety of instruments to measure change in communication skills. Furthermore, the working environment in these maternity wards is well characterised by long hours, crowded wards, high volume of patients, pressure due to organisational hierarchy at hospital and tension in work relationships between residents, nurses and

**Table 4** Numbers (and percentages) of negative responses for items on the observational checklist concerning communication between doctor and woman during labour/delivery, unless otherwise stated

| | Control period N=565 | Intervention period N=631 | | Control period N=565 | Intervention period N=631 |
|---|---|---|---|---|---|
| Mean (SD) of the average score | 0.34 (0.13) | 0.39 (0.10) | | | |
| Mean (SD) of the average score for complete checklist | 0.34 (0.10) | 0.39 (0.09) | | | |
| Checklist item | | | Checklist item | | |
| Did the doctor identify himself/herself? | 539 (100) | 621 (99) | Was the woman asked any questions at this stage? | 168 (32) | 94 (15) |
| Did the doctor call the woman by her name? | 132 (24) | 99 (16) | Did the woman ask any questions at this stage? | 292 (56) | 281 (46) |
| Did the doctor take the woman's permission to examine her? | 326 (61) | 305 (49) | Did the doctor respond to her queries? | 275 (63) | 342 (64) |
| Did the doctor explain the plan he/she will follow? | 497 (92) | 577 (92) | Was the doctor responsive to the woman's pain? | 285 (59) | 333 (57) |
| Before starting the vaginal examination, did the doctor | | | Was the woman encouraged at this stage? | 186 (40) | 239 (42) |
| Take her permission for the exam? | 335 (62) | 262 (42) | Was the woman told about the proximity of labour? | 242 (55) | 363 (66) |
| Close the door? | 472 (88) | 568 (91) | Was the woman told about the stages of labour? | 399 (95) | 533 (98) |
| Cover the woman during the exam? | 315 (59) | 116 (18) | Was the woman told about her role during labour? | 347 (83) | 508 (94) |
| Ask the woman to bend her knees? | 210 (39) | 79 (13) | During delivery, was the woman asked to push? | 26 (6) | 23 (4) |
| After completing the vaginal examination, did the doctor | | | Did they explain to the woman when and how to push? | 237 (57) | 230 (42) |
| Tell the woman the results of the exam? | 418 (78) | 504 (81) | Did the woman ask any questions at this stage? | 284 (68) | 296 (55) |
| Explain the next steps to be followed? | 486 (91) | 586 (94) | Did the doctor respond to her queries? | 264 (77) | 330 (72) |
| Relay the findings to other team members? | 241 (45) | 102 (16) | Did the doctor inform the mother about her baby's status? | 307 (75) | 386 (72) |
| Relay the findings to the midwives/nurses? | 393 (80) | 579 (93) | Was the woman congratulated after the delivery? | 217 (53) | 214 (40) |
| Did the doctor give instructions about eating and drinking? | 463 (87) | 516 (83) | Was the woman instructed about the process after the delivery? | 392 (95) | 515 (96) |
| Did the doctor give instructions about movement? | 335 (63) | 428 (70) | Was the woman instructed about how to care for her newborn? | 409 (99) | 534 (99) |

Owing to missing information some of the percentages do not tie up with the total number given. The columns compare the numbers all the intervention clusters to the control clusters.

midwives.[22] All those factors could have left our target group of residents under pressure, but more importantly could have left women feeling unsatisfied. A recent article by Berridge et al[23] highlighted how poor communication in the delivery suites can compromise safe and efficient care and humane relationships.

The interpretation of the findings should also consider the women's status in a community in which the expectations of the women are very low but also where culture and gender issues are critical. Previous research findings showed that some means of communication, for example, eye-to-eye communication, are not acceptable if gender of the provider is different from that of woman.[12] We also argue that the overall positive satisfaction of women in our observation can be due to the childbirth event by itself as being a pleasing experience as well as getting the service free of charge in public hospitals by those women who come from a poor socioeconomic class.[12] It was surprising to observe that the very positive evaluation of the training sessions by the trainees was not translated to improved satisfaction of the women. However, we fully support Bingham[24] who stressed the importance of carrying out patient satisfaction surveys to help hospitals change or improve their childbirth policy.

Our study enrolled all public teaching maternity hospitals in Damascus. Therefore, our findings can be generalisable to overcrowded public delivery settings in Syria and to settings in developing countries with similar characteristics of overcrowdedness, over worked residents who are not trained in good communication skills and where the role of nurses and midwives is marginal. These findings suggest that training residents only without a structural change in the system and without further reinforcement might not be sufficient to improve women's satisfaction with the service provided.

## Strengths and limitations

The key strength of the study was in its stepped wedge cluster randomised design characterised by being ethical and practical,[16] which resulted in the training package being delivered to all residents serving at the hospital, thereby complementing their medical training in communication skills that is missing from their curriculum. Furthermore, the study had a 100% response rate among women but there were some missing values on some of the questions. High response rate is very common in our culture. Conducting the trial in a developing country certainly contributes to the field as compared with the literature that comes mainly from developed countries.[1–3 9] The primary limitation of the study was its inability to link women's satisfaction, or dissatisfaction, with the behaviour of a specific doctor or specific doctors; for ethical and practical reasons, we observed actions rather than the behaviours of individual residents. This is unavoidable due to the design's key considerations that all hospitals would receive the intervention eventually. Residents are the sole providers of care at labour and delivery in the teaching public

hospitals in our setting and midwives are only marginally involved in care. Therefore, exclusions of nurses and midwives in this study should not be a major limitation in the current setting. The relatively short duration of the training and the lack of follow-up within the hospital setting could have contributed to the negative findings in this study. Furthermore, the selection of public hospitals only in this study might have impacted the generalisability of the findings to other settings; however, this setting is common in our region.[25]

## CONCLUSIONS

In the context of a highly crowded and stressful environment where middle-class and low-class Syrian women deliver, a specially designed training package in interpersonal and communication skills for residents did not achieve an overall improvement in women's satisfaction with doctor–woman relationship in labour and delivery rooms. However, certain items in doctors' behaviour have improved. It would be worth investigating whether the package would improve women's satisfaction in less stressful settings, but also it is worth looking at other possible interventions in maternity care practice such as doctor–midwife collaboration or attendance of birth companion in such settings.

Despite the lack of evidence from this study, the need to improve interpersonal skills of medical doctors and obstetricians specifically should be reinforced, as good communication is central to quality healthcare.[26]

**Acknowledgements** The authors would like to thank Mrs Hoda Hallab who worked with the team to develop the training package and deliver the training at inception. They are also grateful to the directors of the hospitals for facilitating the field work. The effort of Nick Bashour in reviewing the English language of this article is highly appreciated. This article is part of a larger regional research project on Choices and Challenges in Changing Childbirth in the Arab Region sponsored by the Center for Research on Population and Health at the American University of Beirut, Lebanon, with generous support from the Wellcome Trust (UK)(Grant Ref: 074986).

**Contributors** HNB and AAA developed the study concept and aims then initiated the project. MHK, MAT, SAC and MK assisted further in the refinement of the concept and development of the protocol. HNB, MHK and AAA oversaw collection of data. MK did the statistical analysis. HNB and MK drafted the manuscript. All authors read and approved the final version of the manuscript.

**Funding** Wellcome Trust through Faculty of Health Sciences, American University of Beirut.

**Competing interests** None.

**Patient consent** Obtained.

**Ethics approval** Damascus University IRB.

**Provenance and peer review** Not commissioned; externally peer reviewed.

**Data sharing statement** Data set should be available for researchers mainly interested in the stepped wedge cluster randomised trials from Hyam Bashour (email: hbashour@scs-net.org) after written agreements.

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
