## [Reviewer comments · BMJ Open]

Some articles will have been accepted based in part or entirely on reviews undertaken for other BMJ Group journals. These will be reproduced where possible.

ARTICLE DETAILS

TITLE (PROVISIONAL)	The effect of training doctors in communication skills on women's satisfaction with doctor-woman relationship during labour and delivery: a stepped wedge cluster randomised trial in Damascus
AUTHORS	Bashour, Hyam; Kanaan, Mona; Kharouf, Mayada; Abdulsalam, Asmaa; Tabbaa, Mohammed; Cheikha, Salah

VERSION 1 - REVIEW

REVIEWER	Jocelyn DeJong Professor Faculty of Health Sciences American University of Beirut Beirut Lebanon The Wellcome Trust grant under which the research reported here in Syria was funded was received by my institution, the Faculty of Health Sciences, American University of Beirut but before I joined the institution. I currently coordinate the regional network (Egypt, Lebanon, occupied Palestinian territories and Syria) of researchers under which this work was done but I had no direct involvement in the design, conduct or analysis of the study or in the writing up of the paper. This is the first time I review the paper.
REVIEW RETURNED	07-Mar-2013

THE STUDY	English needs careful editing and rewriting in some points for clarity.
RESULTS & CONCLUSIONS	More elaboration is needed both on the context of maternal health care in Syria as noted in the report attached below, and of the limitations of the study. The latter go a long way to explain possibly the lack of positive findings, and are instructive to others doing studies in this area.
REPORTING & ETHICS	Careful attention to ethics.
GENERAL COMMENTS	This paper reports on a Cluster Randomized Trial in Damascus using a stepped wedge design that tested that effectiveness of an intervention to improve health –care providers' communication skills with female patients during labour and delivery in Syria. As I noted when I accepted this review, I am familiar with the work of the research team which is part of a broader regional research network that I coordinate. I am therefore familiar with this study but have not been involved in it and feel I can comment objectively on its merits. I would note at the outset that in the Arab region generally there have been relatively few (and only recently) randomized trials attempting to test the effectiveness of interventions to improve the quality of maternal health care. This is particularly the case in Syria.

Therefore in that respect the study makes an important contribution and studies such as this one should be published not only to provide the evidence needed (and ideas for further research) but also as a role model for other such studies.

I am somewhat familiar with the public health/maternity context in public hospitals in Syria and I can attest that even before the political troubles currently plaguing the country, there was much need for improvement in the quality of maternal health care. The paper could be strengthened in my view by describing more at the beginning of the paper the overall organization of maternity care in the country and its deficiencies, that have been the subject of some research. Particularly, the overcrowded nature of the public hospitals with high patient populations and the delegation of much of the care given to women to residents (as well as the limited role of midwives) should be elaborated on. Moreover the issue, only appearing later in the paper, about the lack of pain relief given to women should be elaborated on earlier in the paper. Some of these issues come out in the discussion in the paper in order to interpret the findings, but prior description would give the reader more of an idea of the context so that they can understand more the significance of the study.

In principle, it is as important to report on negative findings as positive findings and I hope the journal will not be subject to a publication bias that often prevails whereby only positive findings are reported. That said, more interpretation of the limitations of the study and possible explanations for the negative findings are warranted in the paper. One clear deficiency in the design (but which the authors point out was necessary for ethical reasons) was that although individual providers attended the communication intervention training, the assessment was done at the level of shifts rather than following individual providers through successive shifts. That is, there is the possibility that doctors who were not trained were providing care and their communication behavior observed/assessed. As the authors report: "We have chosen this approach as we were more interested in the change of the behavior at the service level rather than the actual change in the behavior of the individual resident."

A second important limitation is that the intervention did not extend to other cadres of health-care workers such as nurses. This limitation is also mentioned but could be further developed, as it could have important implications for other studies. There is also the issue of the duration of the training and the lack of follow-up within the hospital setting. Finally, the important limitation about pain relief was discussed above but again could have been elaborated on.

Some of these lessons about why the intervention did not have the hypothesized impact could be provided in the 'key messages' of the paper as they are instructive to other studies.

	Overall the study raises questions as to whether training individuals without more structural changes in the delivery of care can have an impact on improving the quality of doctor-patient communication. These are important questions to address and call for further research. In addition to the above concerns, I would suggest that the paper be reviewed by a native English language speaker. Some minor comments are listed below:  • Abstract: two thousand women (not thousands) • The first two sentences repeat each – both are about increasing attention – but need to say more centrally why communication is critical to maternal health care • The statement on p. 6 that “doctors and midwives themselves were sources of dissatisfaction” needs to be further explained • Top of p. 9 – break the sentence up to explain first high attendance and then positive evaluations. • On p. 11 the denominators for the numbers of residents for each hospital would be useful (for example 85 out of a total of how many residents? Are these all the residents in that hospital?) • On p. 11, the Al Galaa study in Egypt should be explained and referenced • P. 15 – line four from bottom “the proportion of no was consistently lower” needs to say the proportion of negative responses • Acknowledgements – training at inception rather than kick-off
--	--

REVIEWER	Yana Vinogradova Research Statistician University of Nottingham United Kingdom
REVIEW RETURNED	16-Apr-2013

THE STUDY	Abstract: The last sentence in Results seems neither clear nor accurate. Introduction: The description of the rationale for the study is well written. It is not clear, however, here what was the secondary objective and what observational list the authors refer to. Methods: Line 13, page 7 “...four study hospitals (clusters)...” makes a reader assume that hospitals formed the clusters, but later text refers to “observing 10 clusters” and later – in the same paragraph – “each cluster (hospital)...” adds to the confusion. This should be reworded to define a cluster clearly. Statistical analysis: The authors used Generalized Linear Mixed Model, which is an extension to Linear Mixed Model for non-normal data, but there is no mention in the text about the distribution of the main outcome measure. Further, the authors applied all three types
------------------	--

	of estimates suggested by referenced paper [12] – I would expect the authors to choose a method and justify their choice of the model instead of overwhelming the reader with 3 different estimates. It is also not clear how the observational check list was analysed.
RESULTS & CONCLUSIONS	The tables (and graphs) provide information for each time period instead of giving overall information and telling the reader that the tests did not show significant differences between the time periods. The existing level of detail would be better placed in appendices or left as 'available from the authors'. Reporting Mean (SD) and Median (IQR) for the same variable is also unnecessary, appropriate statistics should be those appropriate to variable distribution. Apart from a long table describing the frequencies for the observational check list no analyses are reported.
REPORTING & ETHICS	The study period, not just the years of training, should be reported. It is also not clear how the randomisation was performed. There is no statement about the generalisability of the findings.
GENERAL COMMENTS	At present the paper contains too many details and is overlong. The statistical methods include formulae for the model, which are obvious for statisticians but complex and unnecessary for physicians. Applying three different types of estimates also weakens it, creating the feel of a 'fishing expedition'. I think shortening the paper to create a more concise manuscript will significantly improve it.

VERSION 1 – AUTHOR RESPONSE

Reviewer: Jocelyn DeJong
Professor
Faculty of Health Sciences
American University of Beirut
Beirut
Lebanon

The Wellcome Trust grant under which the research reported here in Syria was funded was received by my institution, the Faculty of Health Sciences, American University of Beirut but before I joined the institution. I currently coordinate the regional network (Egypt, Lebanon, occupied Palestinian territories and Syria) of researchers under which this work was done but I had no direct involvement in the design, conduct or analysis of the study or in the writing up of the paper. This is the first time I review the paper.

English needs careful editing and rewriting in some points for clarity.

Response: We have edited the paper and rephrased some sections for clarity.

More elaboration is needed both on the context of maternal health care in Syria as noted in the report attached below, and of the limitations of the study. The latter goes a long way to explain possibly the lack of positive findings, and are instructive to others doing studies in this area.

Response: we further elaborated on the context of maternal health care in Syria in the introduction. As for the limitations we expanded on these in the discussion and added the limitations to the article

summary.

This paper reports on a Cluster Randomized Trial in Damascus using a stepped wedge design that tested the effectiveness of an intervention to improve health –care providers' communication skills with female patients during labour and delivery in Syria. As I noted when I accepted this review, I am familiar with the work of the research team which is part of a broader regional research network that I coordinate. I am therefore familiar with this study but have not been involved in it and feel I can comment objectively on its merits.

I would note at the outset that in the Arab region generally there have been relatively few (and only recently) randomized trials attempting to test the effectiveness of interventions to improve the quality of maternal health care. This is particularly the case in Syria. Therefore in that respect the study makes an important contribution and studies such as this one should be published not only to provide the evidence needed (and ideas for further research) but also as a role model for other such studies. I am somewhat familiar with the public health/maternity context in public hospitals in Syria and I can attest that even before the political troubles currently plaguing the country, there was much need for improvement in the quality of maternal health care.

The paper could be strengthened in my view by describing more at the beginning of the paper the overall organization of maternity care in the country and its deficiencies, that have been the subject of some research. Particularly, the overcrowded nature of the public hospitals with high patient populations and the delegation of much of the care given to women to residents (as well as the limited role of midwives) should be elaborated on. Moreover the issue, only appearing later in the paper, about the lack of pain relief given to women should be elaborated on earlier in the paper. Some of these issues come out in the discussion in the paper in order to interpret the findings, but prior description would give the reader more of an idea of the context so that they can understand more the significance of the study.

Response: We have expanded on the introduction to address the points raised by the reviewer. We now provide further information with respect to overcrowding, residents roles and pain relief.

In principle, it is as important to report on negative findings as positive findings and I hope the journal will not be subject to a publication bias that often prevails whereby only positive findings are reported. That said, more interpretation of the limitations of the study and possible explanations for the negative findings are warranted in the paper. One clear deficiency in the design (but which the authors point out was necessary for ethical reasons) was that although individual providers attended the communication intervention training, the assessment was done at the level of shifts rather than following individual providers through successive shifts. That is, there is the possibility that doctors who were not trained were providing care and their communication behavior observed/assessed. As the authors report: "We have chosen this approach as we were more interested in the change of the behavior at the service level rather than the actual change in the behavior of the individual resident."

Response: We now make it clear that all the residents at the study hospitals received the training.

A second important limitation is that the intervention did not extend to other cadres of health-care workers such as nurses. This limitation is also mentioned but could be further developed, as it could have important implications for other studies. There is also the issue of the duration of the training and the lack of follow-up within the hospital setting. Finally, the important limitation about pain relief was discussed above but again could have been elaborated on.

Some of these lessons about why the intervention did not have the hypothesized impact could be provided in the 'key messages' of the paper as they are instructive to other studies.

Response: We now provide the limitations in the “Key messages” and elaborate a bit more on the role of midwives and nurses. We also mention the pain relief policy in the introduction.

Overall the study raises questions as to whether training individuals without more structural changes in the delivery of care can have an impact on improving the quality of doctor-patient communication. These are important questions to address and call for further research.

In addition to the above concerns, I would suggest that the paper be reviewed by a native English language speaker.

Some minor comments are listed below:

- Abstract: two thousand women (not thousands)

Response: done.

- The first two sentences repeat each – both are about increasing attention – but need to say more centrally why communication is critical to maternal health care

Response: we have deleted the repetition in the first two sentences.

- The statement on p. 6 that “doctors and midwives themselves were sources of dissatisfaction” needs to be further explained

Response: we now explain what the sources of dissatisfaction were on Page 6 lines 14 to 15.

- Top of p. 9 – break the sentence up to explain first high attendance and then positive evaluations.

Response: we now report the attendance rate and then comment on the feedback on Page 7 lines 7 to 10.

- On p. 11 the denominators for the numbers of residents for each hospital would be useful (for example 85 out of a total of how many residents? Are these all the residents in that hospital?)

Response: The numbers reflect the total number of residents attending labour and delivery rooms at each hospital. We have reworded how we report the numbers under sample size on Page 13 lines 1 to 3.

- On p. 11, the Al Galaa study in Egypt should be explained and referenced

Response: we now reference and explain a bit further the Al-Galaa study on p.12 lines 7 to 11 (p. 11 previously) and also mention it earlier on in the background.

- P. 15 – line four from bottom “the proportion of no was consistently lower” needs to say the proportion of negative responses

Response: we have amended this phrase in this instance and throughout the manuscript.

- Acknowledgements – training at inception rather than kick-off

Response: Amended as requested.

Reviewer: Yana Vinogradova
Research Statistician
University of Nottingham
United Kingdom

Abstract: The last sentence in Results seems neither clear nor accurate.

Response: we have amended this sentence by changing the currently reported 0.08 to the actual - 0.08, the conclusions reported now are aligned with the CI estimates.

Introduction: The description of the rationale for the study is well written. It is not clear, however, here what was the secondary objective and what observational list the authors refer to.

Response: we have amended the last paragraph of the introduction to explain further the secondary objective on Page 7 lines 1 to 3.

Methods: Line 13, page 7 "...four study hospitals (clusters)..." makes a reader assume that hospitals formed the clusters, but later text refers to "observing 10 clusters" and later – in the same paragraph – "each cluster (hospital)..." adds to the confusion. This should be reworded to define a cluster clearly.

Response: we have amended the description under the study design to clarify what is meant by a cluster and added the participants' flowchart. Please see also our reply to point 4 above.

Statistical analysis: The authors used Generalized Linear Mixed Model, which is an extension to Linear Mixed Model for non-normal data, but there is no mention in the text about the distribution of the main outcome measure. Further, the authors applied all three types of estimates suggested by referenced paper [12] – I would expect the authors to choose a method and justify their choice of the model instead of overwhelming the reader with 3 different estimates. It is also not clear how the observational check list was analysed.

Response: we have amended the statistical analysis to take into account the reviewer's comment and added a sentence to make it clear how the checklist was analysed. Furthermore, we clarified in the results section the analysis of the secondary outcome. We have reported estimates based on Generalized Linear Mixed Model to account for deviation from normality in our data; we now supply a graphical representation of the outcome by intervention arm in the appendix.

The tables (and graphs) provide information for each time period instead of giving overall information and telling the reader that the tests did not show significant differences between the time periods. The existing level of detail would be better placed in appendices or left as 'available from the authors'.

Response: we have amended Tables 1 to 5 to provide overall comparison and included the detailed tables in a supplementary file. Furthermore, we merged Tables 3 and 4 to give the new Table 3 and renumbered subsequent tables.

Reporting Mean (SD) and Median (IQR) for the same variable is also unnecessary, appropriate statistics should be those appropriate to variable distribution.

Apart from a long table describing the frequencies for the observational check list no analyses are

reported.

Response: we have amended Table1 to provide the mean and SD only removed the other statistics as the distribution was not extremely skewed. We have further clarified the analysis that we carried with regard to the observational checklist.

The study period, not just the years of training, should be reported. It is also not clear how the randomisation was performed.

Response: we have now included further information with regard to the study period. Data collection ran from April 2008 to January 2009. We've also provided further information with regard to the randomisation under the Study design.

There is no statement about the generalisability of the findings.

Response: we have added two sentences in the discussion about generalisability of the results on Page 19 lines 3 to 6.

At present the paper contains too many details and is overlong. The statistical methods include formulae for the model, which are obvious for statisticians but complex and unnecessary for physicians. Applying three different types of estimates also weakens it, creating the feel of a 'fishing expedition'. I think shortening the paper to create a more concise manuscript will significantly improve it.

Response: We have deleted from the manuscript the formulae and presented analyses based on the GLMM estimates as these take into account the distribution of the outcome which is currently presented in Figure 2 in the Appendix.

VERSION 2 – REVIEW

REVIEWER	Jocelyn DeJong Professor Faculty of Health Sciences American University of Beirut Beirut Lebanon The Wellcome Trust grant under which this research was funded was received by my institution, the Faculty of Health Sciences, American University of Beirut but before I joined the institution. I coordinate the regional network entitled Choices and Challenges in Changing Childbirth (Egypt, Lebanon, occupied Palestinian territories and Syria) of researchers under which this work was done but I had no direct involvement in the design, conduct or analysis of the study or in the writing up of the paper. This is the first time I review the paper.
REVIEW RETURNED	19-Jun-2013

THE STUDY	The outcome measure needs to be consistently described - on page 4 it is cited as "women's satisfaction with doctor-woman relationship in labour and delivery" and "women's satisfaction with labour and delivery." It needs to be worded in exactly the same way throughout
------------------	--

	the paper. The fourth point on the article focus is not needed and could be replaced by a statement that there have been few RCTs testing interventions to improve maternal health care in the Arab region and in Syria in particular. On page 15, the fact that doctors insulted patients and screamed at them in both the control and intervention arms needs to be addressed in the discussion. There remain some errors in the English and some sentences have awkward wording (e.g. second to last sentence p. 4). The first sentence of the background should clarify that it is health providers' communication skills. There is an error top line of p. 14 - should be patient-doctor not patient-woman relationship. The word 'servants' top of p. 18 should be replaced and clarified. The paper needs to be proofread by a native English speaker.
GENERAL COMMENTS	This is a much improved draft. The outcome measure needs to be consistently described - on page 4 it is cited as "women's satisfaction with doctor-woman relationship in labour and delivery" and "women's satisfaction with labour and delivery." It needs to be worded in exactly the same way throughout the paper. The fourth point on the article focus is not needed and could be replaced by a statement that there have been few RCTs testing interventions to improve maternal health care in the Arab region and in Syria in particular. On page 15, the fact that doctors insulted patients and screamed at them in both the control and intervention arms needs to be addressed in the discussion. There remain some errors in the English and some sentences have awkward wording (e.g. second to last sentence p. 4). The first sentence of the background should clarify that it is health providers' communication skills. There is an error top line of p. 14 - should be patient-doctor not patient-woman relationship. The word 'servants' top of p. 18 should be replaced and clarified. The paper needs to be proofread by a native English speaker.

VERSION 2 – AUTHOR RESPONSE

The outcome measure needs to be consistently described - on page 4 it is cited as "women's satisfaction with doctor-woman relationship in labour and delivery" and "women's satisfaction with labour and delivery." It needs to be worded in exactly the same way throughout the paper.

This was double checked and corrected.

The fourth point on the article focus is not needed and could be replaced by a statement that there have been few RCTs testing interventions to improve maternal health care in the Arab region and in Syria in particular.

Replacement of the statement was done.

On page 15, the fact that doctors insulted patients and screamed at them in both the control and intervention arms needs to be addressed in the discussion.

This point was addressed in the discussion.

There remain some errors in the English and some sentences have awkward wording (e.g. second to last sentence p. 4).

This was reviewed with the aim that it reads better.

The first sentence of the background should clarify that it is health providers' communication skills.

Done

There is an error top line of p. 14 - should be patient-doctor not patient-woman relationship.

Done

The word 'servants' top of p. 18 should be replaced and clarified.

The word was replaced with a more specific one "Cleaners"

The paper needs to be proofread by a native English speaker.

A native speaker was consulted to improve the English. He was acknowledged for better English, hopefully.

As for sharing the data, yes we are willing to do so.